# Temporal transcriptomics suggest that twin-peaking genes reset the clock

William G Pembroke, Arran Babbs, Kay E Davies\*, Chris P Ponting\*, Peter L Oliver\*

MRC Functional Genomics Unit, Department of Physiology Anatomy and Genetics, University of Oxford, Oxford, United Kingdom

**Abstract** The mammalian suprachiasmatic nucleus (SCN) drives daily rhythmic behavior and physiology, yet a detailed understanding of its coordinated transcriptional programmes is lacking. To reveal the finer details of circadian variation in the mammalian SCN transcriptome we combined laser-capture microdissection (LCM) and RNA-seq over a 24 hr light / dark cycle. We show that 7-times more genes exhibited a classic sinusoidal expression signature than previously observed in the SCN. Another group of 766 genes unexpectedly peaked twice, near both the start and end of the dark phase; this twin-peaking group is significantly enriched for synaptic transmission genes that are crucial for light-induced phase shifting of the circadian clock. 341 intergenic non-coding RNAs, together with novel exons of annotated protein-coding genes, including *Cry1*, also show specific circadian expression variation. Overall, our data provide an important chronobiological resource (www.wgpembroke.com/shiny/SCNseq/) and allow us to propose that transcriptional timing in the SCN is gating clock resetting mechanisms.

**\*For correspondence:** kay.davies@dpag.ox.ac.uk (KED); Chris.Ponting@dpag.ox.ac.uk (CPP); peter.oliver@dpag.ox.ac.uk (PLO)

**Competing interests:** The authors declare that no competing interests exist.

## Introduction

The suprachiasmatic nucleus (SCN) of the hypothalamus is the seat of the principal circadian clock in mammals. Entrained by photic information and other external stimuli, this group of neurons maintains 24-hr rhythms of physiology and behaviour by synchronising molecular oscillators in the brain and peripheral tissues (*Welsh et al., 2010*; *Ko and Takahashi, 2006*; *Maywood et al., 2007*). The transcriptional output of the SCN is thus critical for this fundamental biological adaptation to night and day (*Panda et al., 2002*; *Zhang et al., 2014b*; *Li et al., 2015*).

An early microarray study of the SCN identified 650 genes displaying circadian oscillations (*Panda et al., 2002*). However, this number will have been limited by the number of genes surveyed, the low signal-to-noise ratio of certain genes (e.g. *Per2*), and the use of experimental samples likely to include a large proportion of the surrounding hypothalamic tissue. The study of such heterogeneous samples will result in reduced amplitudes of oscillations for genes that cycle asynchronously in different brain regions (*Zhang et al., 2014b*). A recent study estimated that over half of all protein-coding genes show circadian oscillations in the mouse, although only 642 could be identified in hypothalamic samples (*Zhang et al., 2014b*). Furthermore, many circadian transcriptomic surveys of mammalian systems have utilised constant lighting conditions to focus on the endogenous, free-running circadian clock and have not assessed more physiologically relevant light/dark (LD) cycles (*Zhang et al., 2014b*; *Panda et al., 2002*; *Hughes et al., 2009*; *Hurley et al., 2014*).

To identify circadian processes driven by photic cues, we combined RNA-seq with laser-capture microdissection (LCM) of the SCN to provide a comprehensive and tissue-specific transcriptomic investigation of the master pacemaker. These data reveal many hundreds of novel cycling transcripts and provide evidence that the temporal variation in a specific group of genes plays a role in modulating light-induced phase resetting of the circadian clock.

**eLife digest** The daily cycles of life in mammals are driven by a small region of the brain called the suprachiasmatic nucleus (or SCN). The SCN receives signals from sunlight and other environmental factors to help coordinate most aspects of daily biological activity and behaviour. To work correctly, it is essential that the SCN switches certain genes on and off at exactly the right time. However, many questions remain over the identity of these genes and how their levels of activity change during a 24-hour period.

When a gene is active (or "being expressed"), it is used as a template to build the molecules of RNA that are needed to make proteins and to help to control how cells work. Pembroke et al. have now sequenced the RNA molecules made in the SCN of mice (which plays the same role as the equivalent human brain region) over a 24-hour period. The mice spent half of each day in the light, and half in the dark. This revealed that the expression levels of over a quarter of all the genes that are found in the SCN fluctuate over a 24-hour period. One particular group of genes peak in activity twice a day; Pembroke et al. suggest that these genes are important for controlling how an animal can adjust its body clock to light.

Further research is now needed to find out which of the newly discovered fluctuating genes play the most important roles in daily activity rhythms, and which might play a part in disease.

## Results and discussion

### RNA-seq of high quality SCN samples identifies novel SCN enriched genes

LCM accurately isolated the entire SCN for transcriptional profiling (*Figure 1—figure supplement 1A*). Pooled dissected tissue from five adult male mice provided one of three replicates for each of six timepoints over a 12:12 LD cycle (ZT2, 6, 10, 14, 18 and 22). Sequencing RNA in each replicate generated up to 133 million paired-end reads of which 45–65% were subsequently mapped to the mouse genome allowing for gene expression level estimation.

Global expression differences were apparent between timepoints, as revealed by principal component analysis (PCA) (*Figure 1—figure supplement 1B*). Of 10 key circadian-clock genes, 8 showed the classic sinusoidal circadian profile ($q < 0.05$) with a zenith and nadir at the expected timepoints (*Figure 1A*); this verifies the temporal expression of our LCM SCN transcriptome dataset and indicates that any sequencing batch effects are minimal. Furthermore, quantitative (q)PCR of the LCM SCN samples showed a high degree of concordance with the RNA-seq expression data (*Figure 1—figure supplement 1C–D*). Our SCN transcriptomes are also separable from those in other brain regions and non-CNS tissues (*Zhang et al., 2014b*; *Brawand et al., 2011*; *Merkin et al., 2012*; *Menet et al., 2012*; *Barbosa-Morais et al., 2012*; *Azzi et al., 2014*) (*Figure 1—figure supplement 2A*). In addition, four key SCN markers (*Avp*, *Vip*, *Grp* and *Six6*) showed a far greater expression enrichment in the SCN over other brain regions (*Figure 1—figure supplement 2B*). In summary, these data provide the first identification of SCN enriched genes either on a genome-wide scale or over a 24-hr period and allow an anatomically precise and comprehensive examination of the SCN transcriptional output over the full day. To allow further exploration into the dataset, we provide a user-friendly web application (www.wgpembroke.com/shiny/SCNseq/).

A total of 146 genes were identified whose expression is highly enriched in the SCN (Materials and methods; analysis of variance (ANOVA), $q < 0.05$) of which 4 out of 4 were confirmed using in situ hybridisation (ISH) (*Figure 1—figure supplement 2C*; *Figure 1C*). These genes included many known SCN markers, as well as others not previously implicated in circadian biology or in SCN function (*Figure 1C*). Their significant enrichments for the Mouse Phenotype term 'abnormal circadian rhythm' ($q = 1.2 \times 10^{-4}$) and for Gene Ontology (GO) annotations relating to signalling pathways ($q < 0.05$; *Figure 1B*) are consistent with the SCN's role as the master pacemaker. Enrichment for the flagellum annotation and G-protein coupled receptor (GPCR) signalling pathways (*Figure 1B*) may reflect the roles of primary cilia as GPCR signalling hubs (*Omori et al., 2015*). These results thus suggest that GPCR-mediated neuropeptide signalling via cilia may be important for SCN function.

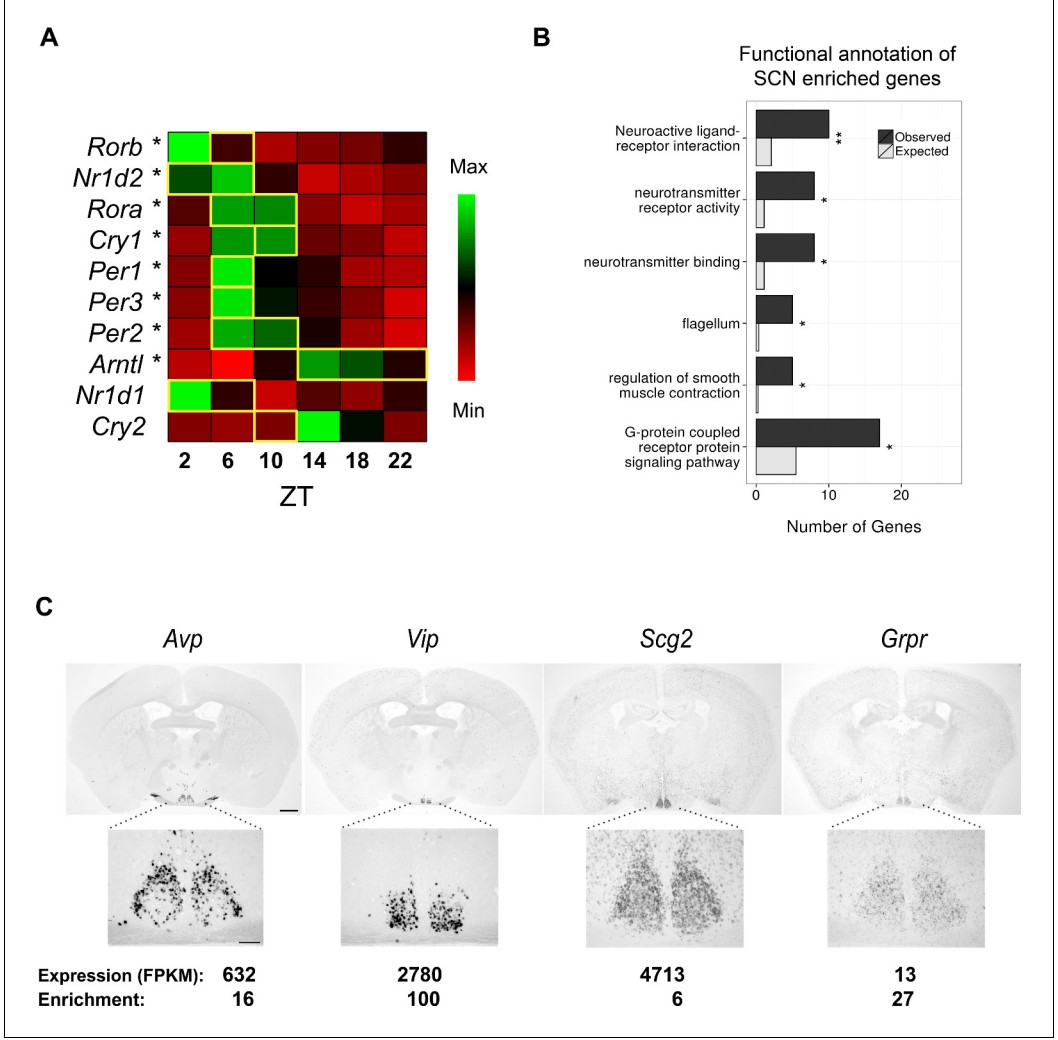

**Figure 1.** RNA-seq data of laser-capture microdissection (LCM) suprachiasmatic nucleus (SCN) samples recapitulate the cycling of clock gene expression and identify known and novel SCN enriched genes. (A) Heatmap showing mean expression of key clock genes in the SCN across circadian time. The majority (8/10) show significant cycling (DESeq2 and JTK Cycle; *q < 0.05) and peak at the expected timepoint denoted by a yellow box (Ko and Takahashi, 2006). (B) Statistically significant Gene Ontology (GO) enrichments for SCN enriched genes (q < 0.05). Bars represent the number of genes of a given category expected by chance among SCN enriched genes (grey) versus those observed (black). q-values: *q < 0.05, **q < 0.01, ***q < 0.001. (C) In situ hybridisation (ISH) of four SCN enriched genes at ZT6 in the wild-type brain. Scale bar: 0.5 mm (low magnification); 0.2 mm (high magnification).

The following figure supplements are available for figure 1:

**Figure supplement 1.** Generation of the SCN transcriptome over 24 hr.

**Figure supplement 2.** Use of publically available RNA-seq data to identify SCN enriched genes.

## Identification of sinusoidally cycling genes

We next identified 4569 genes (24% of 18,889 Ensembl-annotated coding or non-coding models expressed (≥1 read) in over six samples) whose expression oscillated in a sinusoidal manner (*Figure 2—figure supplement 1A,B*) according to both JTK Cycle and DESeq2 approaches (Materials and methods). Importantly, this number of cycling genes is sevenfold higher than previously reported in the SCN (*Panda et al., 2002*). This large increase is likely to reflect a combination of factors: first, the significantly improved anatomical enrichment of the SCN tissue we have surveyed; second, the greater sensitivity and specificity of the statistical approaches we have applied

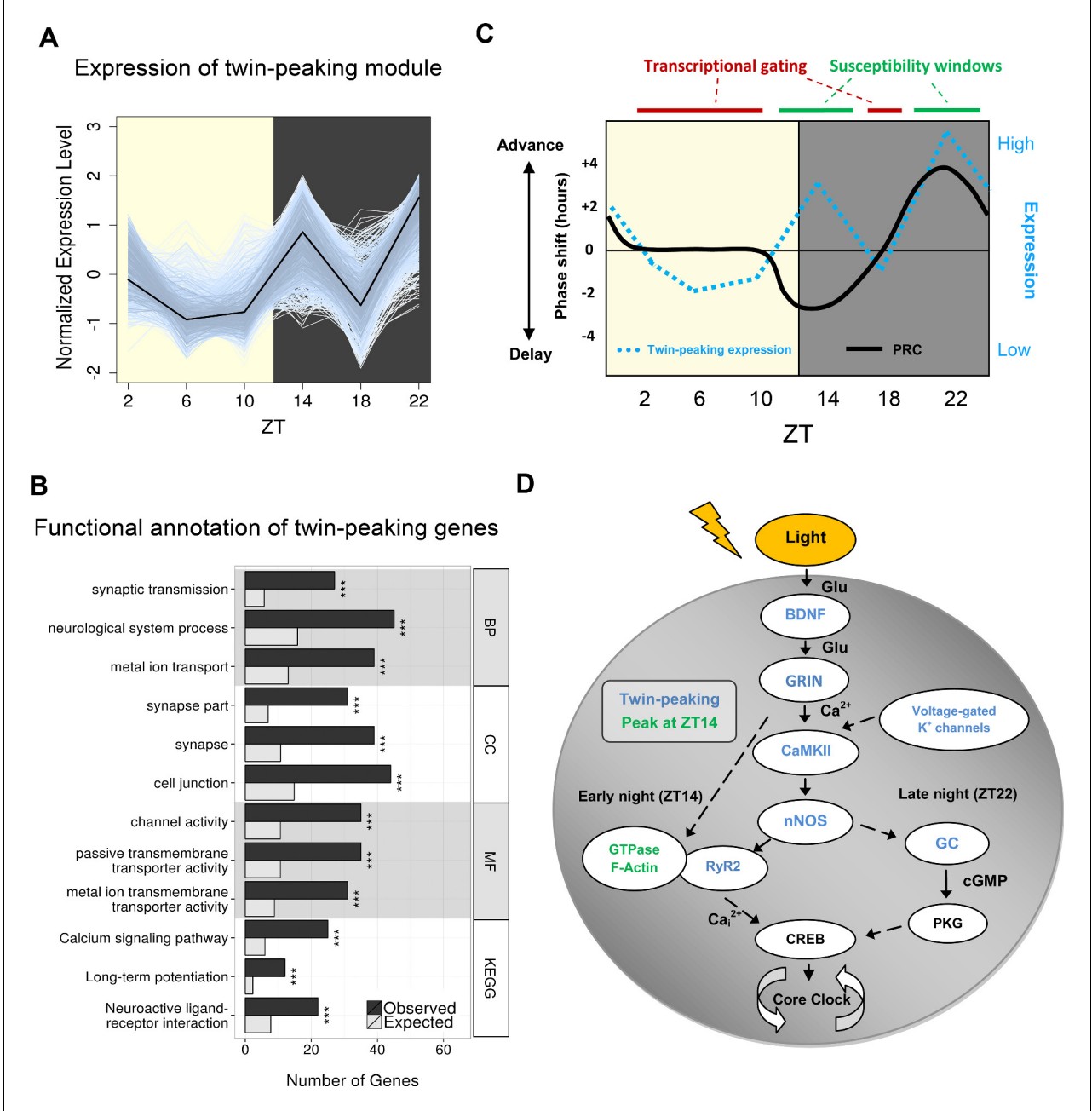

**Figure 2.** Gene clustering identifies a novel twin-peaking synaptic gene module with a potential role in phase resetting. (**A**) Relative expression of each gene present in the identified twin-peaking module over 24 hr. The black line represents the module eigengene – the first principal component of the gene expression matrix for this module. (**B**) Top three statistically significant (q < 0.05) Gene Ontology (GO) (biological process (BP), cellular component (CC), molecular function (MF)) and Kyoto Encyclopedia of Genes and Genomes (KEGG) enriched terms for the 496 significantly fluctuating twin-peaking genes. q-values: *q < 0.05, **q < 0.01, ***q < 0.001. (**C**) Illustration of the theoretical phase response curve (PRC) in mouse, which demonstrates the phase shifting potential when a light pulse is applied at different times (black line) (Golombek and Rosenstein, 2010). Our module eigengene for the twin-peaking module (blue dotted line) indicates that when its expression lies above the threshold level (black horizontal line), the phase of the organism is able to shift. The greater the expression over this threshold, the larger the phase shift. Conversely, when expression is low and below this threshold, the suprachiasmatic nucleus (SCN) is resistant to phase shifting induced by light pulses. This suggests the presence of susceptibility windows (shown by green bars) when light impulses are able to shift the phase of the SCN. (**D**) Circadian gating of light signalling pathway, adapted from (Iyer et al., 2014). Twin-peaking gene-encoded proteins identified by laser-capture microdissection (LCM) SCN RNA-seq are indicated in blue. Photic information is transmitted to the SCN via glutamatergic signalling through the retinohypothalamic tract (RHT). (i) Brain-derived neurotrophic factor (BDNF) gates glutamatergic synaptic transmission by regulating either presynaptic release or membrane channel activity. (ii) Voltage-gated potassium channels hyperpolarise the SCN neurons allowing photic input from the RHT to elicit a significant response. (iii) Glutamate binding to glutamate

*Figure 2 continued on next page*

*Figure 2 continued*

receptor, ionotropic, N-methyl D-aspartate (GRIN) receptors induces calcium influx inducing CaMKII autophosphorylation. (iv) Active pCaMKII phophorylates neuronal nitric oxide synthase (nNOS), which generates nitric oxide (NO), producing differential effects according to time of day. (v) In early night (ZT14), we observe peaking of genes relating to small guanosine triphosphate (GTPase) activation and actin-based cytoskeletal remodelling. At this timepoint, F-actin levels are at their highest and an influx of $Ca^{2+}$ depolymerises the F-actin affecting components of the mitogen-activated protein (MAP) kinase pathway. $Ca^{2+}$ also induces intracellular $Ca^{2+}$ release though the opening of the endoplasmic reticulum calcium channel RyR2. (vi) In late night (ZT22), NO activates guanylyl-cyclase (GC), which increases cyclic guanosine monophosphate (cGMP) levels activating protein kinase G (PKG). (vii) Phosphorylation of cAMP response element-binding protein (CREB) leads to the transcription of *PER1*, which contributes to the shifting of the circadian phase.

The following figure supplement is available for figure 2:

**Figure supplement 1.** Identification of 4569 cycling genes in the SCN.

(*Hughes et al., 2010*); and finally, our sampling approach under LD as opposed to the constant dark (DD) conditions utilised in earlier studies (*Panda et al., 2002*). For example, we are able to detect cycling genes that may be physiologically relevant to light exposure or light-induced transcriptional cascades that are undetectable as cycling under DD (*Ueda et al., 2002*; *Leming et al., 2014*). Our new collection of SCN cycling genes were enriched for GO terms relating to key processes such as translation and mitochondrion-related genes, in line with previous findings (*Figure 2—figure supplement 1C*; *Panda et al., 2002*). De novo transcription has been shown to drive rhythmicity of approximately 20–30% of all cycling genes (*Koike et al., 2012*; *Menet et al., 2012*), which illustrates the importance of post-transcriptional regulation in shaping the circadian transcriptome. Interestingly, we observe novel enrichments among cycling genes in terms relating to RNA splicing and ribonucleoprotein (RNP) complexes (*Figure 2—figure supplement 1C*) which suggests that these processes may establish oscillatory expression patterns (*Wang et al., 2013*). Other post-transcriptional mechanisms, such as miRNA level fluctuation, which were not assessed in this experiment, could also contribute to initiating these patterns.

Approximately half of the SCN cycling genes (2451/4569) showed no evidence for sinusoidal expression in 12 other tissues, including the whole hypothalamus, based on published circadian expression data (JTK Cycle; q < 0.05; *Figure 2—figure supplement 1D*) (*Zhang et al., 2014b*). These genes included those essential for normal SCN function, including *Prok2*, *Rorb* and *Nts*, which were also validated by quantitative polymerase chain reaction (qPCR) (*Figure 1—figure supplement 1C*). This indicates that many genes cycle exclusively in the SCN, or only cycle under a 12:12 LD lighting schedule rather than under the constant dark conditions used previously (*Zhang et al., 2014b*). Comparing our results under LD with published DD datasets may inflate the perceived number of genes cycling solely in the SCN, however, as there are genes likely to cycle under LD exclusively as shown by microarray studies in lower organisms (*Ueda et al., 2002*; *Leming et al., 2014*). Six of the nine genes which cycled significantly across all 12 other tissues (q < 0.05) also cycled in the LCM SCN samples (*Arntl*, *Dbp*, *Nr1d2*, *Per2*, *Per3* and *Tsc22d3*) suggesting a core set of universally cycling genes (*Figure 2—figure supplement 1D*). In support of this notion our LCM SCN cycling gene list significantly overlapped with those cycling in 8 of 12 other tissues (q < 0.05), as well as those cycling in an SCN microarray study (*Zhang et al., 2014b*; *Panda et al., 2002*) (*Figure 2—figure supplement 1E*). As expected, the SCN, hypothalamus and cerebellum showed the greatest overlap of their circadian transcriptomes (*Figure 2—figure supplement 1E*).

## Identification of novel twin-peaking synaptic module

Previous studies only considered circadian gene expression following a sinusoidal pattern. Alternative and potentially functionally important expression profiles may thus have been overlooked. Using Weighted Correlation Network Analysis (WGCNA), which does not assume a single mode of temporal variation, we identified a single module of 766 genes, which instead of adopting a sinusoidal profile, displayed two peaks of expression at ZT14 and ZT22 (*Figure 2A*). From this 'twin-peaking' module 496 genes individually exhibited statistically significant temporal fluctuations (DESeq2, q < 0.05), of which 17 of 17 were validated by qPCR (*Figure 1—figure supplement 1E*). This profile is also distinct from previously described 12-hr harmonics (*Cagampang et al., 1998a*;

*1998b*; *Hughes et al., 2009*), and prior to this, only single genes from individual studies were shown to possess similar expression patterns (*Shinohara et al., 1993*; *Cagampang et al., 1998c*; *1998d*). For example, isoforms of protein kinase C (*Prkcb* and *Prkcg*) exhibit a twin-peaking profile in rat SCN similar to that observed in our RNA-seq data (*Cagampang et al., 1998c*). This twin-peaking transcriptional profile, particularly for this large number of genes, was thus unexpected. Functional annotations related to synaptic transmission, calcium signalling and gated channel activity were greatly and significantly enriched among these twin-peaking genes (q < 0.05; *Figure 2B*), suggesting a circadian component to establishing the electrophysiological environment of the SCN.

The peaking of expression at ZT14 and ZT22 coincides with phase shifting 'susceptibility windows'; time periods in which the photic input permits phase shifting (or 'resetting') of the clock (*Ding et al., 1994*; *Iyer et al., 2014*). This is demonstrated by the phase response curve (PRC): a light impulse at ZT14 will initiate a phase delay of the clock whereas a light impulse at ZT22 will initiate a phase advance (*Figure 2C*) (*Golombek and Rosenstein, 2010*; *Ding et al., 1994*). At all other timepoints we assessed under an LD cycle the twin-peaking genes exhibited low expression levels exactly when the clock is resistant to phase shifts. This suggests a possible transcriptional gating mechanism in which a threshold level of gene expression needs to be exceeded for light to activate this phase shifting pathway (*Ding et al., 1994*; *Gillette and Mitchell, 2002*; *Iyer et al., 2014*). Transcriptional profiling at a higher temporal resolution would be required to confirm the elevated expression of twin-peaking genes above this critical threshold at timepoints when light-induced phase shifts are possible (e.g. ZT12; *Figure 2C*). Genes previously known to participate in this light-induced phase shifting pathway display this twin-peaking pattern (*Figure 2D*), and their pharmacological inhibition (e.g. for *Grin2a/b*, *Camk2a*, *Nos1*, *Ryr2*) blocks light-induced phase shifts (*Ebling et al., 1991*; *Yokota et al., 2001*; *Ding et al., 1998*; *1994*; *2007*; *2010*). Many genes whose expression peaks singly at ZT14 have functions related to actin cytoskeleton reorganisation and/or to small GTPase signalling (*Figure 2—figure supplement 1F–G*). This supports the hypothesis that actin depolymerisation is important for phase delays but not phase advances (*Gerber et al., 2013*; *Cabej, 2013*). Taken together, our data indicate that oscillations in the SCN transcriptome play an important role in gating the differential responses to light, a fundamental circadian process. The complex relationship between photic and non-photic cues and their relative roles in phase adjustment must also be considered, however. Light exposure during constant conditions can still modify the processing of non-photic signals by the SCN (*Challet and Pevet, 2003*). For example, novelty-induced running in the subjective mid-day induces a phase advance in mice, although a 1-hr light pulse administered after access to running wheels can significantly attenuate this affect (*Biello and Mrosovsky, 1995*). More work is required to determine how the timing or relative level of gene expression mediates these light-induced behavioural changes.

## Identification of novel transcripts with circadian expression profiles

A total of 3187 multiexonic long intergenic non-coding RNA (lincRNAs; >200 nucleotides) loci were identified from the SCN transcriptome data. Considered together, they displayed evidence of significant sequence conservation, had lower expression and fewer exons than protein-coding genes in line with lncRNAs identified in other mammalian tissues (*Zhang et al., 2014a*; *Cabili et al., 2011*; *Ponjavic et al., 2007*), but did not display correlated expression with that of the nearest protein-coding gene suggesting these lincRNAs are often not functioning as enhancer (e)RNAs (*Figure 3—figure supplement 1A–D*). Of these lincRNAs, 341 (11%) displayed significant temporal fluctuation, including lincRNA NONCO7761 (*3100003L05Rik*) whose presence in the SCN was confirmed by ISH at ZT6 (*Figure 3D*) as well as by qPCR to peak at both ZT6 and ZT18 (*Figure 3E*). In addition, 28 of these fluctuating lincRNAs peaked at ZT14 and ZT22 (*Figure 3B–C*), and thus may modulate or be modulated by the transcriptional or post-transcriptional regulation of protein-coding genes of the twin-peaking synaptic module described earlier.

We also identified 1013 novel exons – transcribed regions of the transcriptome which have not been annotated by Ensembl or UCSC, but contribute to alternative transcripts in a known gene. Of these, we focused on the novel exon of the core clock gene *Cry1* which appears to represent an alternative transcriptional start site (TSS) (*Figure 4A*). Despite its large size (322 bp) and high expression, the novel exon present in the first intron of *Cry1* had remained elusive, yet is confirmed by reverse transcription polymerase chain reaction (RT-PCR) from SCN tissue (*Figure 4C*). The novel and canonical *Cry1* isoforms are expressed in antiphase, as confirmed by qPCR from LCM SCN

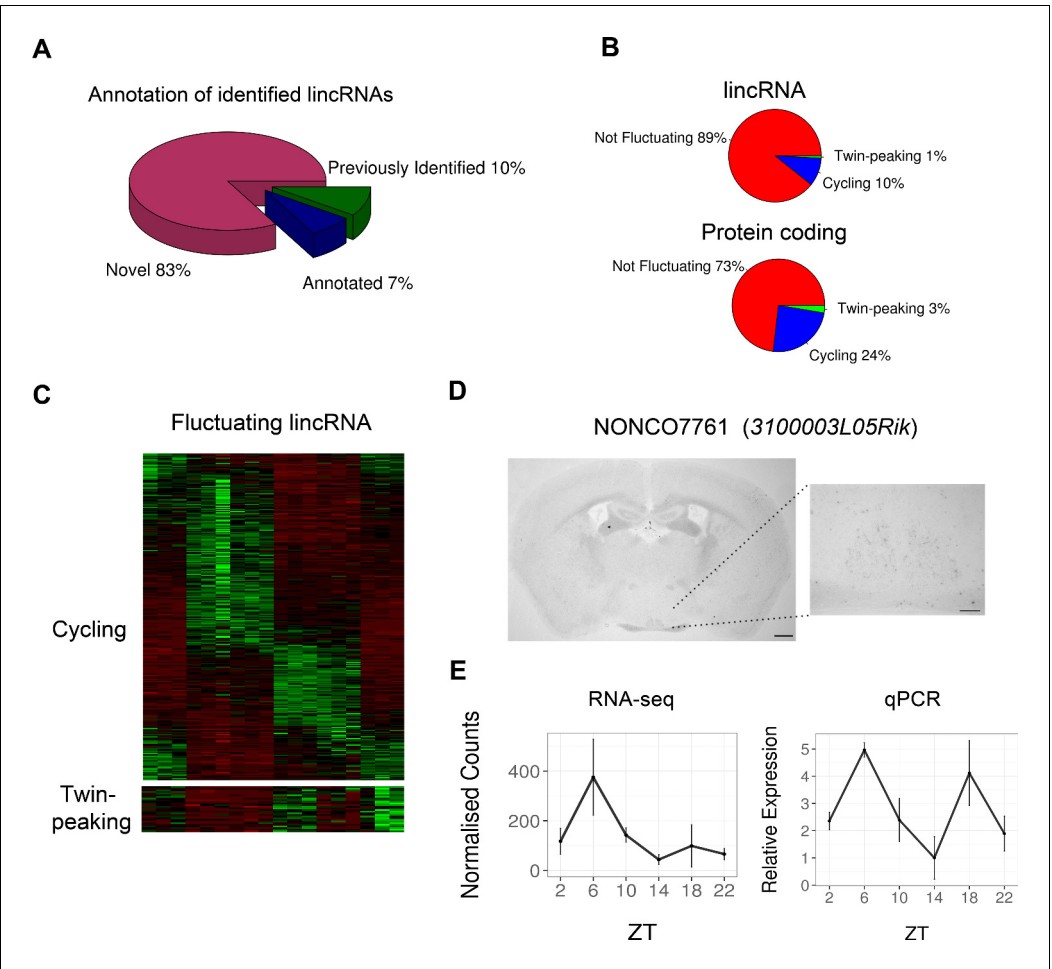

**Figure 3.** Identification of multiexonic lincRNAs whose expression fluctuates in the suprachiasmatic nucleus (SCN). (**A**) Proportion of the 3187 identified lincRNAs which (i) have been previously annotated (according to Ensembl or UCSC), (ii) are not annotated in these resources but have been identified in other studies (Belgard et al., 2011, Ramos et al., 2013) or (iii) are unique to this study. (**B**) Proportion of the 3187 identified lincRNAs which show significant fluctuating expression patterns (twin-peaking module and DESeq2 q < 0.05, or JTK cycle q < 0.05 and DESeq2 q < 0.05); protein-coding genes are shown for comparison. (**C**) Heatmap showing the expression levels in each biological replicate for the 341 fluctuating lincRNAs. (**D**) In situ hybridisation (ISH) of NONCO7761 (*3100003L05Rik*) at ZT6 in the mouse brain showing low yet distinct expression in the SCN. Scale bar: 0.5 mm (low magnification), 0.2 mm (high magnification). (**E**) RNA-seq and quantitative polymerase chain reaction (qPCR) show fluctuating mean expression ± SD of the lincRNA NONCO7761 (*3100003L05Rik*) over 24 hr.

The following figure supplement is available for figure 3:

**Figure supplement 1.** Properties of identified SCN lincRNAs.

samples (*Figure 4D*), indicating an unanticipated mode of temporal regulation in the SCN for this core clock gene.

In conclusion, this first temporal analysis of the SCN using RNA-sequencing reveals the presence of a twin-peaking transcriptional module with a suspected function in circadian control, and identifies thousands of novel transcripts, including a novel *Cry1* isoform, which may play important roles in SCN function.

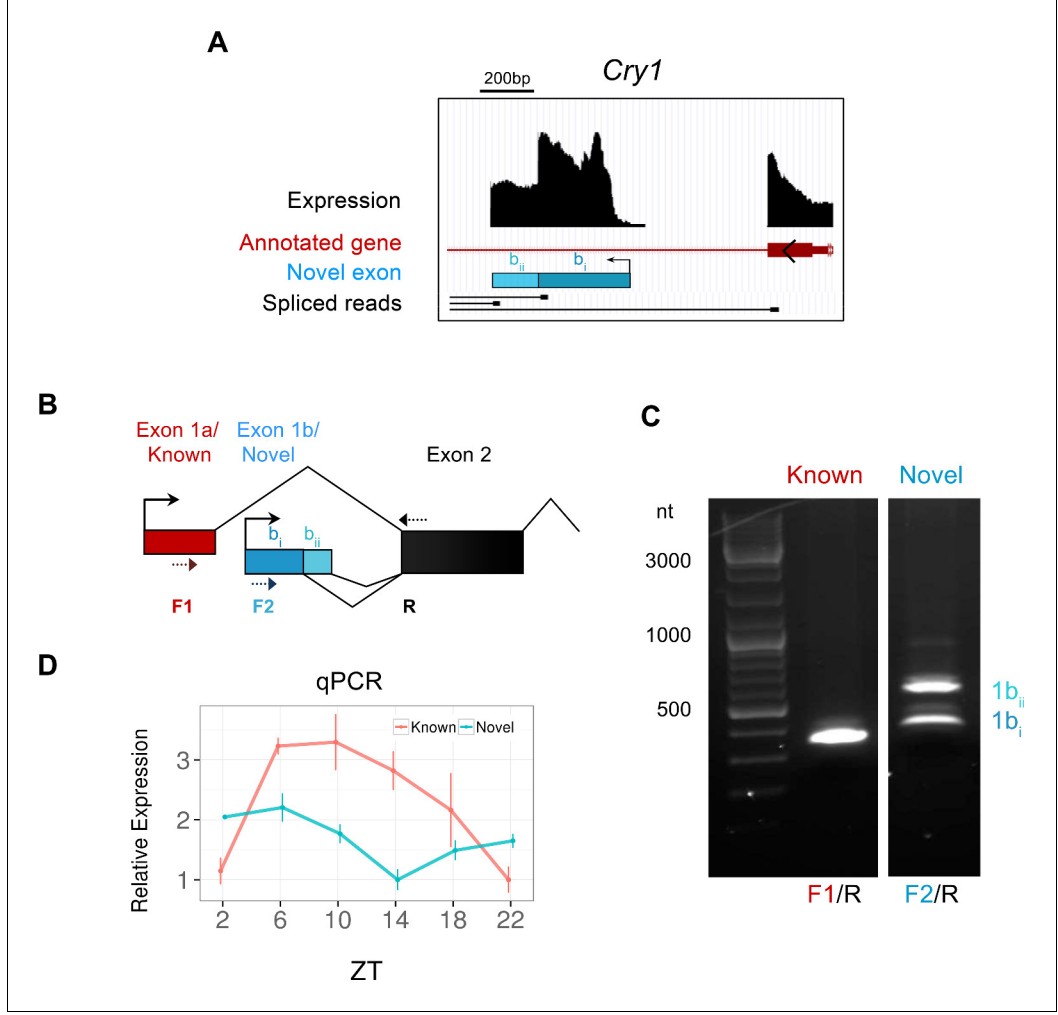

**Figure 4.** Identification of a novel antiphase *Cry1* isoform. (**A**) Genomic region of mouse chromosome 10 displaying the novel exon (blue bar) for the clock gene *Cry1*, which appears to represent an alternate transcriptional start sites (TSS). Peaks (black) signify total expression of the exon whereas representative RNA-seq spliced reads (black) indicate exonic splice junctions. (**B**) Schematic diagram showing the 5′ gene structure of the canonical (red) and novel (blue) *Cry1* isoforms. The novel exon (exon 1b) splices into the subsequent second exon using two different exon boundaries. Dotted arrows denote the position of PCR primers used to amplify *Cry1* by RT-PCR. (**C**) RT-PCR from SCN RNA at ZT6 confirms the presence of novel *Cry1* TSS capable of producing two different isoforms as labelled. The identity of all PCR products was confirmed by sequencing. (**D**) Quantitative polymerase chain reaction (qPCR) of the canonical and novel *Cry1* TSS show antiphase cycling expression (mean ± SD) patterns.

## Materials and methods

### SCN RNA sample preparation

All animal experimentation has been approved by the UK Home Office and the University of Oxford Ethical Review Board. Male C3H/HeH mice at 10–12 weeks of age were group housed in light-tight chambers equipped with light emitting diode (LED) lighting at 150 lx at the cage floor. After 7 days of acclimatisation, mice were singly housed for 7 days prior to tissue harvesting. Brains were removed at one of six Zeitgeber times (ZT) and immediately frozen on dry ice in optimal cutting temperature (OCT) mounting media (VWR, Lutterworth, United Kingdom). Eighteen frozen coronal sections at 15 μM were cut spanning the entire rostral to caudal region of the SCN and mounted onto polyethylene naphthalate (PEN) slides (Carl Zeiss Ltd., Cambridge, United Kingdom). LCM was carried out from dehydrated, Nissl-stained sections using the PALM system (Carl Zeiss Ltd., Cambridge, United Kingdom) as previously described (*Dulneva et al., 2015*). Tissue from five animals was pooled to generate one biological replicate sample (15 mice in total were used per ZT) and total

RNA was purified using the RNeasy micro kit (Qiagen, Manchester, United Kingdom). RNA quality was determined using an RNA Picochip (Agilent, Stockport, United Kingdom), with RNA Integrity Number (RIN) values over 8 for all samples and average yield of approximately 10 ng per replicate. Approximately 1 ng of RNA was amplified using the SMARTer protocol (Takata Clontech Bio Europe SAS, France), and the resulting libraries were 100-bp paired-end sequenced on the HiSeq (Illumina, Fulbourn, United Kingdom; Wellcome Trust Centre for Human Genetics Sequencing Core). The average read pair count obtained was ~35 M per technical replicate (~100 M per biological replicate).

## Acquisition of additional RNA-seq data sets

One SCN, six brain and five liver transcriptome datasets from six independent studies (GSE54124, GSE30352, GSE54652, GSE41637, GSE41338 and GSE36874) were downloaded from the Gene Expression Omnibus (GEO) database to allow comparison of expression between tissues. PCA revealed separation of datasets according to tissue rather than experimental factors (*Figure 1—figure supplement 2A*) suggesting negligible batch effects and therefore differences in expression are likely due to biological, rather than technical differences.

## Read processing and mapping

All sequencing data was processed as follows unless otherwise specified. To retain only high quality reads an in-house script was used to: (i) trim ends of reads with a Phred base quality score below 3, (ii) remove specified over-represented adapter or contaminating sequences, (iii) remove reads with a length below 50 bp (or 40 bp if initial sequencing length was shorter (*Merkin et al., 2012*; *Barbosa-Morais et al., 2012*) and (iv) remove any reads with an average Phred quality score below 30. Reads were multimapped against the Genomic Mapping and Alignment Program for mRNA and EST sequences (GMAP; version 2012-07-20; *Wu and Watanabe, 2005*) and processed mm10/GRCm38.70 reference genome using Genomic Short Nucleotide Alignment Program (GSNAP; *Wu and Nacu, 2010*) with the option to consider novel splice sites. Data are deposited in GEO accession number GSE72095. Further details of all genes surveyed are shown in *Supplementary file 2*.

## Identification of novel exons

De novo transcript assembly was conducted using cufflinks v2.0.2 (*Trapnell et al., 2010*) allowing identification of 6610 novel exons. These were required not to overlap but were within a 10-kb window of any feature in the most recent Ensembl or UCSC gene annotation sets. Of these exons, only those (i) with a reciprocal overlap of <25% with any retroposed pseudogene (ucscRetroAli6), (ii) with a reciprocal overlap of <50% with any transposable element, (iii) with over 20 spliced reads in the novel exon, (iv) with over 10 spliced reads per 100 bp of the novel exon, (v) with over 20 reads that splice into a known transcript and were (vi) shorter than 3000 bp, were retained providing a robust set of 1013 novel exons. Further details of all novel exons surveyed are shown in *Supplementary file 4*.

## Identification of lincRNAs

Identification of lincRNAs was conducted through the use of the CGAT NGS pipelines rnaseqtranscripts.py and rnaseqlncrna.py (*Sims et al., 2014*). The first pipeline identifies transfrags using cufflinks and retains those present in at least two samples. The second predicts lncRNAs by removing transfrags which overlap protein-coding exons. These lncRNAs are then assessed for coding potential using the coding potential calculator (CPC; *Kong et al., 2007*) and removed if annotated as 'coding' (CP score >1). Only the intergenic (>2 kb from any protein-coding gene), multiexonic lncRNAs with expression (≥1 read) in over six biological replicates were chosen for further investigation. Further details of all lincRNAs surveyed are shown in *Supplementary files 3 and 5*.

## Expression quantification

To determine the expression level for each gene in each sample, aligned reads were quantified using either HTSeq (version 0.5.4p1; *Anders et al., 2015*) or cuffNorm. Both methods used a GTF file generated from combining both the most recent Ensembl annotation and the identified set of lincRNAs.

FPKM values were generated using cuffQuant and normalised with cuffNorm, whereas count tables were generated using HTseq and normalised using DESeq2 (*Love et al., 2014*).

## Identification of SCN enriched genes

To identify SCN enriched genes, genes were required to show significantly greater expression (q < 0.05, Benjamini–Hochberg) in the brain than in liver and significantly greater expression (q < 0.05, Benjamini–Hochberg) in both this study's SCN dataset and the published SCN dataset (*Azzi et al., 2014*) relative to the brain using ANOVA in R. FPKM values were then averaged for each study and quantile normalised. Fold-change (FC) enrichment scores were calculated by averaging the FPKM values for each study for a particular tissue and then comparing this average to the mean FPKM value for another tissue. A gene was defined as being SCN enriched if it showed a threefold enrichment in the SCN over brain samples.

## Identification of fluctuating genes

The R packages JTK_CYCLE (v2.1; *Hughes et al., 2010*) and DESeq2 (v 1.6.3) were used together to identify genes whose expression significantly altered over time with a sinusoidal oscillatory pattern. DESeq2 was used to perform a likelihood ratio test to determine the genes whose expression significantly altered over time (q < 0.05, Benjamini–Hochberg) and the JTK_CYCLE was used to identify genes whose expression followed 24-hr periodic waveforms (q < 0.05, Bonferroni). To identify genes with non-sinusoidal oscillatory patterns, signed WGCNA (*Langfelder and Horvath, 2008*) was used to detect modules based upon gene coexpression. Modules were merged if their similarity was greater than 0.3 according to dendrogram height. Of the resultant 23 modules only the 'light-steelblue1' module was investigated further because it was the only non-sinusoidal module whose expression pattern was not influenced by anomalous expression of a single sample at a particular timepoint. Genes of this module were defined as fluctuating if their expression significantly altered over time (q < 0.05, Benjamini–Hochberg; DESeq2).

## Gene-annotation enrichment analysis

Enrichment analysis was performed using the Database for Annotation, Visualization and Integrated Discovery (DAVID) Functional Annotation Tool v6.7 (Huang da et al., 2009). DAVID incorporates Fisher's exact test and was used to determine significant (q < 0.05, Benjamini) GO terms and Kyoto Encyclopedia of Genes and Genomes (KEGG) pathways. GO terms were passed through REVIGO to prune redundant terms with a semantic similarity of 0.5, 0.7 or 0.9, depending on the length of the GO term (*Supek et al., 2011*). Mammalian Phenotype (MP) enrichment was determined using the Mouse Genome Informatics (MGI) database (*Eppig et al., 2015*).

## qRT-PCR and RT-PCR

For validation of results from RNA-seq, qRT-PCR was performed using the BioMark system (Fluidigm) as according to the two-step single-cell gene expression protocol using EvaGreen as described in the Real-Time PCR Analysis User Guide (PN 68000088, Fluidigm). For the process, 666 ng of starting RNA was used for each sample, which underwent 18 cycles of specific target amplification (STA) followed by a tenfold dilution. SCN CT values were normalised using the housekeeping genes *ActB*, *Gapdh* and eight genes which showed the most stable expression in the SCN according to sequencing data (*Ankrd40*, *Nkiras1*, *Sar1a*, *Smarce1*, *Snrpn*, *Tpgs2*, *Tsn* and *Ywhaz*). Primer sequences are listed in *Supplementary file 1*.

For RT-PCR of novel exons of *Cry1*, total RNA was purified from dissected SCN tissue and used to prepare cDNA (Revertaid, Fermentas). Primers representing previously annotated (5' GTGAGGAGGTTTTCTTGGAAG 3') and novel (5' CTTCTAGGGAATTGCGACTG 3') exons were used with a common reverse primer (5' CTGGGAAATACATCAGCTGG 3') to amplify products by RT-PCR followed by visualisation on an agarose gel and sequencing. The genomic coordinates of the novel exon are: chr10: 85,183,342 to 85,183,832 (mm10) with splicing into the following exon at 85,183,342 or 85,183,510.

## In situ hybridisation

Brains from male mice representing ZT6 were isolated as above and frozen sections at 15 μM were cut and mounted onto Superfrost slides (VWR). Regions of the selected transcripts were cloned into pCR4-TOPO (Life Technologies, Paisley, United Kingdom) for DIG-labelled riboprobe synthesis: *Avp* (1–490 bp of accession number NM_009732.1), *Vip* (191–715 bp of NM_011702.1), *Scg2* (844–1287 bp of NM_009129.1), *Grpr* (1287–1741 bp of NM_008177.1) and NONCO7761 (*3100003L05Rik;* 32–583 bp of NR_045907). Slide hybridisation and riboprobe detection was carried out as described previously (*Chodroff et al., 2010*).

## Acknowledgements

We thank Adam Handel for technical assistance with the Fluidigm Biomark system, Sergei Maslau for sharing code used to process sequencing data and members of the Ponting group, in particular, Wilfried Haerty, Harry Clifford and T. Grant Belgard, for helpful discussions. We also thank Lorna Witty and the WTCHG genomics core for assistance with the sequencing, Ben Edwards for mouse tissue collection and Russell Foster and Stuart Peirson for valuable scientific input. This work was supported by the UK Medical Research Council and the Sleep and Circadian Neuroscience Institute (SCNi), a Wellcome Trust Strategic Award.

## Additional information

### Funding

| Funder | Grant reference number | Author |
| --- | --- | --- |
| Medical Research Council | Programme Grant, Functional Genomics Unit | Kay E Davies Chris P Ponting |
| Wellcome Trust | Sleep and Circadian Neuroscience Institute (SCNi) | Kay E Davies |

The funders had no role in study design, data collection and interpretation, or the decision to submit the work for publication.

### Author contributions

WGP, PLO, Conception and design, Acquisition of data, Analysis and interpretation of data, Drafting or revising the article; AB, Conception and design, Acquisition of data; KED, Conception and design, Drafting or revising the article; CPP, Conception and design, Analysis and interpretation of data, Drafting or revising the article

### Author ORCIDs

Chris P Ponting, http://orcid.org/0000-0003-0202-7816

### Ethics

Animal experimentation: All experiments were conducted in adherence to the guidelines set forth by the UK Home Office regulations under Project Licence number 30/2792, and with the approval of the University of Oxford Ethical Review Board.

## Additional files

### Supplementary files

• Supplementary file 1. Primer list for quantitative polymerase chain reaction (qPCR).

• Supplementary file 2. Table with statistics and general information for each gene surveyed in the SCN.

• Supplementary file 3. Table with statistics and general information for each multi-exonic lincRNA expressed in the SCN.

• Supplementary file 4. Table with coordinates and additional information for identified novel exons expressed in the SCN.

• Supplementary file 5. A GTF file of the 3187 multi-exonic lincRNA expressed in the SCN.

## Major datasets

The following datasets were generated:

| Author(s) | Year | Dataset title | Dataset URL | Database, license, and accessibility information |
|---|---|---|---|---|
| William G Pembroke, Arran Babbs, Kay E Davies, Chris P Ponting, Peter L Oliver | 2015 | Data from: Temporal transcriptomics suggest that twin-peaking genes reset the clock. | http://dx.doi.org/10.5061/dryad.s5q88 | Available at Dryad Digital Repository under a CC0 Public Domain Dedication. |
| William G Pembroke, Arran Babbs, Kay E Davies, Chris P Ponting, Peter L Oliver | 2015 | Temporal transcriptomics suggest that twin-peaking genes reset the clock | http://www.ncbi.nlm.nih.gov/geo/query/acc.cgi?acc=GSE72095 | Publicly available at the NCBI Gene Expression Omnibus GSE3823 (Accession on: GSE72095). |

The following previously published datasets were used:

| Author(s) | Year | Dataset title | Dataset URL | Database, license, and accessibility information |
|---|---|---|---|---|
| Brown S, Azzi A | 2014 | Circadian behavior is light-reprogrammed by plastic DNA methylation (sequencing) | http://www.ncbi.nlm.nih.gov/geo/query/acc.cgi?acc=GSE54124 | Publicly available at the NCBI Gene Expression Omnibus GSE3823 (Accession on: GSE54124). |
| Brawand D, Soumillon M, Necsulea A, Julien P, Csárdi G, Harrigan P, Weier M, Liechti A, Aximu-Petri A, Kircher M, Albert FW, Zeller U, Khaitovich P, Grützner F, Bergmann S, Nielsen R, Pääbo S, Kaessmann H | 2011 | The evolution of gene expression levels in mammalian organs | http://www.ncbi.nlm.nih.gov/geo/query/acc.cgi?acc=GSE30352 | Publicly available at the NCBI Gene Expression Omnibus GSE3823 (Accession on: GSE30352). |
| Zhang R, Lahens NF, Ballance HI, Hughes ME, Hogenesch JB | 2014 | A circadian gene expression atlas in mammals: Implications for biology and medicine | http://www.ncbi.nlm.nih.gov/geo/query/acc.cgi?acc=GSE54652 | Publicly available at the NCBI Gene Expression Omnibus GSE3823 (Accession on: GSE54652). |
| Merkin JJ, Burge CB | 2012 | Evolutionary dynamics of gene and isoform regulation in mammalian tissues | http://www.ncbi.nlm.nih.gov/geo/query/acc.cgi?acc=GSE41637 | Publicly available at the NCBI Gene Expression Omnibus GSE3823 (Accession on: GSE41637). |
| Barbosa-Morais NL, Kutter C, Watt S, Odom DT, Blencowe BJ | 2012 | The evolutionary landscape of alternative splicing in vertebrate species | http://www.ncbi.nlm.nih.gov/geo/query/acc.cgi?acc=GSE41338 | Publicly available at the NCBI Gene Expression Omnibus GSE3823 (Accession on: GSE41338). |

Menet JS, Rodri-guez J, Rosbash M | 2012 | Nascent-Seq Reveals Novel Features of Mouse Circadian Transcriptional Regulation [ChIP-seq] | http://www.ncbi.nlm.nih.gov/geo/query/acc.cgi?acc=GSE36874 | Publicly available at the NCBI Gene Expression Omnibus GSE3823 (Accession on: GSE36874).

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
