## [Decision Letter]

Thank you for submitting your work entitled "Temporal transcriptomics suggest that twin-peaking genes reset the clock" for consideration by *eLife*. Your article has been reviewed by two peer reviewers, and the evaluation has been overseen by Louis Ptáček (Reviewing Editor) and Aviv Regev as the Senior Editor.

The reviewers have discussed the reviews with one another and the Reviewing editor has drafted this decision to help you prepare a revised submission.

Summary:

In this submission (manuscript and on-line tool), Pembroke and colleagues report on a transcriptomic profile and search tool for the suprachiasmatic nuclei (SCN), site of the brain's main circadian pacemaker. They improve on earlier studies of this type by 1) limiting the sampling of the tissue to the SCN itself (some other studies did not have this degree of neuroanatomical resolution and instead sample the SCN along with extraSCN tissue). 2) by sampling under the physiologically relevant light-dark cycle (as opposed to constant lighting). By combining RNAseq with 1 and 2, they provide exciting new evidence that many other transcripts cycle in a SCN specific manner as do long intergenic non-coding RNAs. Further, they show that some transcripts show two peaks in expression per 24h. Additionally they characterise a novel isoform of the key clock gene, *Cry1*.

They use established algorithms and analytic tools to identify functionally related groups of genes whose expression cycles and confirm these with qPCR In some instances they also confirm expression via in situ hybridisation. The manuscript is linked to an on-line tool which appears relatively easy to use and allows researchers to interrogate this database. This is a valuable resource, particularly for researchers with limited knowledge/insight into these types of investigations.

Essential revisions:

1) In their description of the times of rising and falling transcripts, the authors highlight that the peak in some transcripts occurs at time when light cannot shift mouse rhythms and while others peak at times when light can shift the mouse SCN clock. There is somewhat of an over interpretation in this aspect (paragraph two, subheading “Identification of novel twin-peaking synaptic module”) since mice light-pulsed mice in constant dark will show shifts during the late subjective day or early subjective night. The authors show this is in Figure 2. Indeed, light pulses during the day can counteract shifts to non-photic stimuli given at these times. This means that the apparent paucity/abundance of transcript may or may not be important for shifts to light. This aspect needs to be softened in the text.

2) When using the on-line resource, it is clear that the level of variability of some transcripts can be very high. For example, for *npy2r* and *avpr1a*, the variability in night (ZT18 or 22) is much higher than at other time points. The authors need to comment on why this is the way it is and also describe further the functional consequences.

3) In Figure 1, is it GRP receptor transcript or GRP transcript itself that was measured?

4) For the transcripts showing 2 peaks in expression per 24h, this pattern has been previously observed (Cagampang et al., 1998a, b) and this merits discussion.

5) For reporting replicates, could the authors please provide details on the total number of mice used – is it 5 mice=1 data so samples from 15 mice are combined for the average expression profile at a given time point?

6) As the authors themselves point out, the fact that they find 7x more oscillating transcripts than Panda 2002 is simply a reflection of improved methodologies, or the fact that their experiments were conducted in LD rather than DD conditions. Some might also arise due to more evolved statistical criteria. Discussion on this point is warranted.

7) The claim that half the genes oscillating in the SCN do not oscillate in other tissues is unfortunately marred by the fact that they are comparing their L:D dataset to D:D datasets, as the authors state. This should be discussed.

---

## [Author Response]

Essential revisions:

*1) In their description of the times of rising and falling transcripts, the authors highlight that the peak in some transcripts occurs at time when light cannot shift mouse rhythms and while others peak at times when light can shift the mouse SCN clock. There is somewhat of an over interpretation in this aspect (paragraph two, subheading “Identification of novel twin-peaking synaptic module”) since mice light-pulsed mice in constant dark will show shifts during the late subjective day or early subjective night. The authors show this is in Figure 2. Indeed, light pulses during the day can counteract shifts to non-photic stimuli given at these times. This means that the apparent paucity/abundance of transcript may or may not be important for shifts to light. This aspect needs to be softened in the text.*

The highest levels of expression of the twin-peaking genes we identified are correlated with well-established susceptibility windows for phase-shifting in rodents. However, as the reviewers correctly state, light is also able to induce shifts at times when our twin-peaking genes do not ‘peak’ (such as at the end of the light phase); in these cases, we suggest that mRNA abundance remains above a particular threshold to facilitate phase shifting, despite not peaking at this time point. To illustrate this hypothesis more clearly, we have amended Figure 2 so that when the expression of the twin-peaking module lies above a threshold level (the horizontal line) it allows the potential for a light-induced phase shift.

In addition, we agree that there are undoubtedly complex interactions between photic and non-photic cues that are important for phase adjustment behaviour, and we have now cited additional examples and expanded upon this point in the main text (paragraph two, subheading “Identification of novel twin-peaking synaptic module”). We do not believe that these specific studies necessarily affect our more general hypothesis that transcript abundance is important for light-responsive phase shifting. Nevertheless, we have toned-down our argument in several places (e.g. paragraph two, subheading “Identification of novel twin-peaking synaptic module” and subheading “Identification of novel transcripts with circadian expression profiles”) by acknowledging that more work will be required to determine the link between transcript levels and light-mediated behavioural changes.

*2) When using the on-line resource, it is clear that the level of variability of some transcripts can be very high. For example, for npy2r and avpr1a, the variability in night (ZT18 or 22) is much higher than at other time points. The authors need to comment on why this is the way it is and also describe further the functional consequences.*

Our experimental design sought to reduce technical and biological variation in the data by pooling tissue from 5 animals for each individual sample and counterbalancing for factors such as the position of the cage in the light-controlled chamber prior to dissection. We also re-assessed potentially confounding factors such as RNA quality (RIN value) and the number of sequencing reads/library size obtained. None of these factors was found to be biased towards certain time points or time of day.

Although the two genes mentioned above do show large variation at these time points, they are not representative of the transcriptome as a whole. For example, the related gene *Avp* shows a very small degree of variation in expression at the same time points despite low levels of expression towards the end of the dark phase. In addition, by examining the coefficient of variance (CV) for all genes at each time point, we found that there was not a trend for greater variance of expression during the night as suggested, although it was marginally elevated during the day (12% increase; p<10^-5^ Kruskal-Wallis test).

Nevertheless, the reviewers raise an interesting point that increased variability at certain times could have a functional consequence. For example, a gene with an elevated CV at a particular time point may not need to be tightly regulated at that specific time; expression may only be required to reach a threshold level to induce its necessary function, with the generation of additional mRNA having no detrimental outcome. Variation may also be dampened at the functional level due to translational efficiency or other molecular feedback mechanisms. Alternatively, a gene might have a reduced CV representing tighter regulation at a particular time point if the precise level of mRNA at that time is functionally important.

In summary, from our own RNA-seq data we are reluctant at this stage to come to biological conclusions regarding the variation of single genes based on three biological replicates at each time point, but we are confident that there is no time-dependent technical bias in the dataset that influences our conclusions.

*3) In Figure 1, is it GRP receptor transcript or GRP transcript itself that was measured?*

*Grpr* encodes the GRP receptor.

*4) For the transcripts showing 2 peaks in expression per 24h, this pattern has been previously observed (Cagampang et al., 1998a, b) and this merits discussion.*

In the text we had mentioned 12-hour harmonics, but we were not aware of these particular studies highlighting ‘twin-peaking’ genes as shown by in situ hybridisation, including isoforms of PKC. We thank the reviewers for this observation and have cited these studies in the text (paragraph one, subheading “Identification of novel twin-peaking synaptic module”).

*5) For reporting replicates, could the authors please provide details on the total number of mice used – is it 5 mice=1 data so samples from 15 mice are combined for the average expression profile at a given time point?*

This is correct; we have now further clarified this point in the Methods (subheading “SCN RNA sample preparation”).

*6) As the authors themselves point out, the fact that they find 7x more oscillating transcripts than Panda 2002 is simply a reflection of improved methodologies, or the fact that their experiments were conducted in LD rather than DD conditions. Some might also arise due to more evolved statistical criteria. Discussion on this point is warranted. 7) The claim that half the genes oscillating in the SCN do not oscillate in other tissues is unfortunately marred by the fact that they are comparing their L:D dataset to D:D datasets, as the authors state. This should be discussed.*

We have added additional discussion and citations to address both these important points (paragraphs one and two, subheading “Identification of sinusoidally cycling genes”).